# Increased consumption of ultra-processed foods and worse diet quality in colorectal cancer patients after colostomy: A prospective study

**Arenamoline Xavier Duarte[1]ᵒ, Karine de Almeida Silva[1]ᵒ, Isabela Borges Ferreira[1]‡, Cristiana Araújo Gontijo[2]‡, Geórgia das Graças Pena[1,2]ᵒ ***

1 Graduate Program in Health Sciences, Federal University of Uberlandia, Uberlandia, Minas Gerais, Brazil,
2 School of Medicine, Nutrition Course, Federal University of Uberlandia, Uberlandia, Minas Gerais, Brazil

ᵒ These authors contributed equally to this work.
‡ IBF and CAG also contributed equally to this work.
* georgia@ufu.br

## Abstract

Colorectal cancer (CRC) is commonly treated with intestinal resections that lead to colostomy, which can influence changes in eating habits. This study aimed to analyze energy and nutrient intake, diet quality, and food consumption based on the processing level in CRC patients after colostomy. A prospective study was carried out at three time points (T0–recent colostomy, T1–3 months after colostomy, and T2-6 months after colostomy). Food intake was assessed by 24-hour dietary recall. Macro-micronutrient consumption, the Brazilian Healthy Eating Index-Revised (BHEI-R), and food consumption according to processing level by *NOVA* classification (raw or minimally processed, processed, and ultra-processed foods) were estimated. Generalized estimating equations were used to compare the food intake variables with time points. Of the 46 patients, 52.2% were women, and the mean age was 60.6±12.2 years old. There was a change in food consumption over time, with an increase in energy consumption (kcal and kcal/kg), lipids, and sodium, in addition to a reduction in some nutrients such as protein (g and g/kg), fiber, vitamin B1 and C and phosphorus. Regarding the key outcomes, BHEI-R and NOVA classification showed a poor diet quality with a reduction in total index (p = 0.022), raw food (p = 0.001), total fruits, and whole fruit consumption (p = 0.001), and an increase in sodium (p = 0.001) at 3 and/or 6 months after colostomy concomitant an increase in ultra-processed food (p = 0.015). Nutritional counseling is essential in care, effective eating changes habits improvement of symptoms and nutritional status, besides avoiding potential cancer recurrence.

## Introduction

Colorectal cancer (CRC) is a public health problem with a global incidence of 1,926,425 cases, and 904,019 related deaths were reported in 2022 [1]. CRC treatment may require

**Data Availability Statement:** All relevant data are within the manuscript and its Supporting Information files.

**Funding:** This study was financed in part (publication fee) by the Coordenação de Aperfeiçoamento de Pessoal de Nível Superior - Brasil (CAPES) - Finance Code 001. The funder had no role in study design, data collection and analysis, publication decision, or manuscript preparation.

**Competing interests:** The authors have declared that no competing interests exist.

chemotherapy and/or radiotherapy, intestinal resection and ostomy insertion [2]. Annually, approximately 100,000 ostomy surgeries are performed in the United States [3].

Ostomy due to changes in intestinal function requires some management during the different adaptations, considering the underlying pathology, and one of the management options is dietary change, which aims to control intestinal function before some gastrointestinal symptoms. [4–6]. Commonly documented, patients can present a higher gut transit time with semi-liquid stools and/or constipation, which negatively impacts the patient's quality of life. These conditions can also influence the food consumption changes. Furthermore, when constipation occurs, it is also common to observe increases in ostomy prolapse risk, abdominal discomfort, and pain; these factors lead to reduced food consumption as well as some nutritional deficiencies [4–10].

Many studies have investigated dietary risk factors for developing CRC, such as excess consumption of red meat or processed meats, alcohol intake, and ultra-processed foods [9–11]. However, studies are scarce involving food consumption in CRC patients, especially studies that consider the impact of colostomy on food intake [4, 9–16]. Studies on food consumption have investigated only macro- and micronutrients in a quantitative way [4, 7–12]. In this sense, it is partially known the metabolic demand of the disease can cause depletion of some nutrients such as vitamins D, E and C, selenium, zinc, iron, folic acid, and electrolytes, due to reduced digestion and absorption capacity and disturbed body homeostasis. Furthermore, radiotherapy can cause symptoms and also lead to changes in the intestinal mucosa, influencing nutrient malabsorption and intestinal disorders, such as diarrhea, which can persist for up to ten years after treatment [17]. The surgery increases metabolic energy expenditure, requiring higher energy and nutrients to repair homeostasis, recovery, convalescence, and rehabilitation [18]. Therefore, the demand for nutrients and treatment side effects require nutritional support. Similarly, eating habits, for example, can also be influenced by treatment and after surgery. Antineoplastic treatment can cause some changes in taste and lack of appetite [19]. In the postoperative period, some challenges can be faced, such as nausea, lack of appetite, opioid-induced constipation, and lack of information to optimize the diet [20]. However, a healthy diet and better nutritional monitoring contribute to the improvement of symptoms and nutritional status. The study by Gigic *et al.* (2017) showed that patients with CRC after 12 months of surgery who had a healthy eating pattern, which included greater consumption of fruits and vegetables, reduced some symptoms such as diarrhea [21]. In the study by Ravelli *et al* (2020), patients with CRC who had nutritional monitoring preoperatively and up to 3 months after surgery showed recovery of nutritional status, high food consumption, and reduction of symptoms (abdominal distension, abdominal pain, and early satiety) [22]. However, studies have not explored beyond the nutrients consumed or management of symptoms during the treatment, as diet quality and ultra-processed food intake that are recognized cancer risk factors.

The few studies that focused on diet quality also aimed to investigate the incidence of CRC [11, 14–16], and those that investigated diet quality in CRC survivors did not examine patients with ostomies [23–25]. Furthermore, studies have investigated food intake according to the level of food processing and the development of chronic non-communicable diseases risk [26]. Alternatively, studies that have assessed CRC [10, 11] do not investigate the impact of treatment or colostomy on food consumption. To the best of our knowledge, this is the first study to assess food intake variables such as diet quality and food consumption according to the processing level after colostomy.

So, the relevance of nutritional monitoring of cancer patients is associated with immediate and long-term treatment, reducing possible nutritional deficiencies, increasing post-surgical recovery, minimizing complications, and reducing hospital costs [18, 22].

Through the dearth of information related to food, and clinical implications (or repercussions) that can modify the food consumption of this public, the present study aimed to analyze the energy and nutrient intake, diet quality, and changes in food consumption based on the processing food levels in CRC patients after a recent colostomy.

## Materials and methods

### Participants, study design, and ethical aspects

A prospective study was carried out from August 3rd 2017 to May 13th 2019 in a reference service in a tertiary University Clinical Hospital in Minas Gerais, Brazil. This hospital service is public and financed by Brazil's unified health system (Sistema Único de Saúde [SUS]). This hospital offers inpatient and outpatient health care, and it is a reference for medium and high complexity for neighboring municipalities and it is considered the third largest university hospital in the country. The participants were CRC patients who recently underwent a colostomy at baseline and followed up at 3 and 6 months after the surgical ostomy. The Human Research Ethics Committee of the Federal University of Uberlandia approved this study (protocol number CAAE 65975817.6.0000.5152 and 2.062.182) and all participants signed a free and informed consent form.

### Eligibility criteria

The inclusion criteria were as follows: aged 18 years or older; either sex; CRC diagnosis; and recently underwent a colostomy, regardless of stage. An ostomy was considered recent when the time between the surgery date and the first interview was less than 2 months. The exclusion criteria were as follows: a previous diagnosis of severe depression, neuropsychopathies or other serious mental illness registered in medical records, or chronic diseases whose nutritional recommendations lead to important changes in food consumption.

### Data collection

The data collection occurred at three time points: T0 (0–2 months after the ostomy), T1 (3 months after the T0), and T2 (6 months after the T0). A total of 65 eligible patients were invited. Of these, 17 refused to participate (disinterest (n = 6), displacement (n = 5), assistance in another service (n = 4), *weakness* (n = 1), lack of time (n = 1)), and 2 patients who were not possible to contact. Despite all efforts and attempts, unfortunately, 8 patients in T1 and 7 in T2 did not complete any 24-hour recall (24HR). Therefore, 46 individuals participated in this study, including 14 participants in T0-T1-T2, 23 individuals at T0-T1, and 22 individuals at T0-T2 (Fig 1).

Data for pathological stage (I, II, III, IV or pathological anatomy Y), diagnosis (colon or rectal tumor), treatment (surgery only, chemotherapy or radiotherapy and chemoradiotherapy), and comorbidities (diabetes, arterial hypertension, heart disease, hypothyroidism, Chagas disease, deep vein thrombosis) were collected from clinical records. Sociodemographic data (age, sex, schooling, family income) was based on national questionnaires' criteria [27, 28], anthropometric data (weight and height), data on the use of nutritional therapy, and about the first 24HR, as detailed below were collected by face-to-face interviews at all those time points. Regarding the interviewers, only two registered dietitians assisted by one trained undergraduate student collected all the data. The calibration of each questionnaire was performed before the patient interview.

### Treatment

The treatment protocol for patients with rectal cancer in this reference service is neoadjuvant chemoradiotherapy (5 sessions once a week/sessions with 5-fluorouracil (5-FU) at 500 mg/m$^2$

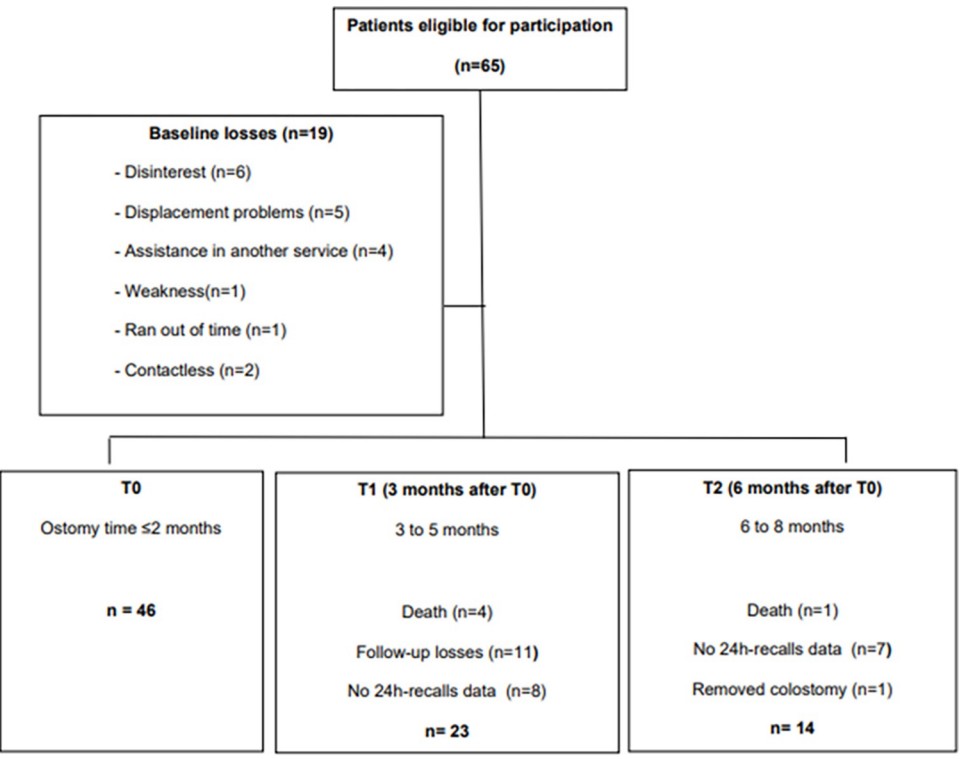

**Fig 1. Diagram reporting the number of patients with colostomy due to colorectal cancer screened and recruited in this study.**

for chemotherapy and 5 sessions of radiotherapy with 50 Gy concomitantly with chemotherapy). The protocol for patients diagnosed with colon cancer is primarily surgery and, in advanced cases, adjuvant chemotherapy. The analyses reported here were adjusted for the treatment.

## Anthropometric measures

Height and body weight were measured according to the World Health Organization (WHO) criteria and used to calculate the body mass index (BMI) [29, 30]. Height and weight were measured using Welmy® scale and stadiometer (São Paulo, Brazil) with precision of 0.1 kg and 0.1cm, respectively. BMI was calculated as body weight (kg)/height (m$^2$), according the WHO reference values for adults and the values indicate the following: following BMI <18.5 underweight; BMI ≥18.5 and <25 normal weight; BMI ≥25 and <30 overweight; and BMI ≥ 30 obesity [29]. The Pan American Health Organization (PAHO) recommends the following classifications for elderly individuals (≥ 60 years old): BMI <23.0 underweight; BMI ≥ 23 and <28 normal weight; BMI ≥ 28 and <30 overweight; and BMI ≥ 30 obesity [31].

## Dietary assessment

Food consumption was assessed through the 24HR at all three time points (T0, T1 and T2). Following the literature, we tried to administer up to three 24HRs for each time point. At T0, 29 patients completed more than one 24HR, but 17 patients completed only a single 24HR. At T1, 13 patients completed only one 24HR, and 10 completed more than one 24HR. At T2, 18

patients completed more than one 24HR, while 4 patients completed a single 24HR. Considering the aim of our study, to assess the food group changes according to the time points and this instrument is capable of estimating the average intake of populations, the application of only one 24HR together with statistical techniques to reduce the effect of within-person variability in the distribution may be sufficient for analysis. Therefore, we considered all patients who completed at least one 24HR. In cases of two 24HRs, the average consumption was calculated [32]. The first 24HR was administered via face-to-face interviews, and subsequent recalls were administered via telephone for up to 30 days [33] following the Multiple Pass Method [34]. The macro- and micronutrient consumption were estimated using the DietPro® Software, version 5.7, with reference, preferably, to the Brazilian Food Composition Table (TACO) and sequentially to the U.S. Department of Agriculture (USDA) [35, 36]. To assess consumption considering energy density, some nutrients, such as fiber; monounsaturated and polyunsaturated fatty acids; cholesterol; vitamins B1, B2, B6 and C; iron, calcium, potassium, phosphorus, magnesium, manganese and zinc were evaluated and normalized to 1000 kcal.

## Diet quality assessment

Diet quality was assessed using the Brazilian Healthy Eating Index-Revised (BHEI-R), an indicator adapted from the Healthy Eating Index (HEI) for the Brazilian population [37]. The BHEI-R is composed of 12 components: total grains (including cereals, roots and tubers); whole grains; total fruit (all fruit including fruits and fruit juice); whole fruit (all fruit excluding fruit juice); total vegetables (including legumes after reaching the maximum for meat, eggs and legumes); dark green and orange vegetables and legumes (including legumes after reaching the maximum for meat, egg and legumes); milk and dairy products (including milk and dairy products as well as soy-based beverages); meats, eggs and legumes; oils (including polyunsaturated fatty acids, proteins, oilseed oils and fish fat); saturated fat; sodium; and SoFAAS (calories from solid fats, alcohol and added sugars) [37].

Consumption measurement units in grams or milliliters were used to calculate the number of servings and, subsequently, the total score for each component in the index. Finally, the number of daily servings was adjusted for energy density (1000 kcal/day). Depending on the component, scores can range from 0 (minimum) to 5, 10 or 20 (maximum) points.

The maximum score was considered for components with consumption equal to or greater than the portions recommended by the Brazilian Dietary Guidelines [38]. For the absent consumption was considered score zero; and for intermediate values of consumption, a calculation was carried out proportional to the quantity consumed. For components such as saturated fat, sodium, and SoFAAS, the proportion is inverse–that is, the higher the intake, the lower the score assigned. The total BHEI-R is the sum of the scores of the components and can reach up to 100 points, with higher scores indicating a healthy diet quality.

## Consumption food according to processing level

The consumption of food according to the processing food level was performed using the *NOVA* classification [39] grouping the consumption into four levels. The first group includes raw or minimally processed food considered by the edible parts of plants or animals or when subjected to processes such as removal of inedible or unwanted parts and with the purpose of increasing the duration of the food, such as pasteurization and freezing, among others. The second group is made up of culinary ingredients, including essential ingredients directly from foods of the first group or from fat. The third group is processed foods, composed of products manufactured with a supplement of some methods (s) belonging to the second group, raw or

minimally processed food, the processes being involved by various methods of preservation and cooking. The fourth group is composed of ultra-processed industrial formulations with five or more ingredients, characterized by the presence of dyes, stabilizers, preservatives, and other food additives [40]. After being classified into the respective groups, the energy value of each group was compared to the energy intake, obtaining a contribution percentage for each group.

## Statistical analysis

Variable distributions were evaluated using the Shapiro–Wilk test. Descriptive statistics were expressed as percentages, means, deviations and/or standard errors. Generalized estimating equation (GEE) models were used to compare food consumption variables at three colostomy time points (T0, T1 and T2). The best distribution model was when the lowest quasilikelihood criterion was found under the independence model. Bonferroni's post hoc test was used to adjust for multiple comparisons. Gender, age (years), type of treatment, family income, and staging were adjusted in the models since they were considered confounding factors. A p-value $\leq 0.05$ was considered statistically significant. All data were analyzed using the Statistical Package for Social Sciences (SPSS), version 25.0 for Windows (SPSS Inc., Chicago, Illinois, United States of America).

To identify if the sample size was sufficient, the effect sizes were estimated based on literature criteria [41, 42] for all variables of the BHEI-R and *NOVA* classification. Therefore, the observation power was estimated by G*Power, version 3.1 (Heinrich-Heine-University Düsseldorf, Düsseldorf, Germany); a sample was considered sufficient when the observation power was equal to or greater than 0.8 [43] (S1 Table). Besides that, to show the follow-up losses did not represent a bias, we performed an analysis comparing the participants who stayed at least two time points with those who withdrew from our study (S2 Table).

## Results

The sociodemographic and clinical data are shown in Table 1. Of the 46 patients at T0, 52.2% were female, and the mean age was 60.6± 12.2 years old. According to clinical data, 70.5% had rectal cancer, 30.4% were in stage III, and 54.3% underwent chemoradiotherapy. Regarding nutritional status, 28.3% were underweight at T0; this prevalence decreased to 13.6% at T2. On the other hand, 19.6% were overweight at T0; this prevalence decreased to 18.2% at T2. Oral nutritional supplementation was used by 21.7% of patients at T0; this prevalence decreased to 18.2% at T2.

Considering macro- and micronutrients (Table 2), a significant increase was observed in total energy consumption (p = 0.001), per kilogram of weight (kcal/kg) (p = 0.009), and lipids (p = 0.019) between T0 and T1. Furthermore, significant decreases were observed in protein (g/kg) (p = 0.041) and phosphorus (p = 0.009) between T1-T2; fiber (p = 0.021) and thiamine (p = 0.013) between T0-T1; monounsaturated fatty acids (p = 0.001) between T0-T2; and vitamin C (p = 0.004) between T0-T2. Sodium showed an increase in consumption over the evaluated time points (p = 0.001). The consumption of other macro- and micronutrients did not differ over time.

Finally, there was a significant reduction in many components in the BHEI-R over time. Significant decreases were observed in the total BHEI-R between T0-T2 (p = 0.022), in total fruit between T0-T1 (p = 0.033), T0-T2 (p = 0.001) and T1 -T2 (p = 0.001), and in total vegetable components in T1-T2 (p = 0.021). The whole fruit component decreased at all time points (p<0.05). In addition, there was an increase in the score of total vegetables at T0-T1 (p = 0.012). On the other hand, there was a significant reduction in the sodium score between

**Table 1. Sociodemographic and clinical data of patients with colostomy due to colorectal cancer.**

| Variables | % (n) |
|---|---|
| **Age (years), mean ± SD** | 60.6 ± 12.2 |
| **Age group** | |
| 19–54 | 15.2 (7) |
| 55–59 | 21.7(10) |
| 60–64 | 26.1(12) |
| 65–69 | 19.6 (9) |
| 70–82 | 17.4 (8) |
| **Sex** | |
| Male | 47.8 (22) |
| Female | 52.2 (24) |
| **Family income (minimum wages)** [*] [£] | |
| < 1 | 8.7 (4) |
| ≥ 1–2 | 26.1 (12) |
| ≥ 2–3 | 30.4 (14) |
| ≥ 3 | 30.4 (14) |
| **Education level (years)** [£] | |
| < 9 | 53.3 (24) |
| ≥ 9–12 | 24.4 (11) |
| ≥ 12 | 22.2 (10) |
| **Clinical Diagnosis** | |
| Colon tumor | 29.5 (13) |
| Rectal tumor | 70.5 (31) |
| **Staging** [£] | |
| I | 21.7 (10) |
| II | 19.6 (9) |
| III | 30.4 (14) |
| IV | 6.5 (3) |
| Pathological anatomy Y | 15.2 (7) |
| **Treatment** | |
| Surgery | 13.0 (6) |
| Chemotherapy or radiation therapy | 32.6 (15) |
| Chemoradiotherapy | 54.3 (25) |
| **Comorbidities** [£] | |
| Diabetes, yes | 13.0 (6) |
| Hypertension, yes | 34.8 (16) |
| Other heart diseases | 8.7 (4) |
| Others diseases, yes | 17.4 (8) |
| **Death** | 8.7 (4) |
| **BMI Classification (kg/m$^2$)** [£] | |
| Underweight | |
| T0 | 28.3 (13) |
| T1 | 30.4 (7) |
| T2 | 13.6 (3) |
| Normal weight | |
| T0 | 50.0 (23) |
| T1 | 39.1 (9) |
| T2 | 50.0 (11) |

(*Continued*)

**Table 1.** (Continued)

| Variables | % (n) |
|---|---|
| Overweight | |
| T0 | 19.6 (9) |
| T1 | 17.4 (4) |
| T2 | 18.2 (4) |
| Oral nutritional supplement | |
| T0 | 21.7 (10) |
| T1 | 17.4 (4) |
| T2 | 18.2 (4) |

* Monthly minimum wage at the time was equivalent to R$998.00.

£ n may vary according to the variability of the data.

¥ Hypothyroidism, Chagas Disease, Deep Vein Thrombosis.

T0 (recent colostomy, n = 46), T1 (between 3 months after T0, n = 23) and T2 (6 months after T0, n = 22).

Abbreviation: BMI: Body Mass Index.

**Table 2. Descriptive analysis of food consumption according to colostomy time of colorectal cancer patients, using generalized estimating equations model.**

| Energy and nutrients | T0 (n = 46) | T1 (n = 23) | T2 (n = 22) | p-value |
|---|---|---|---|---|
| | Mean ± Standard Error | | | |
| **Energy (kcal/day)** | 1460.43 ± 119.91[a] | 2137.37 ± 220.95[a] | 1595.45 ± 233.86 | **0.001** |
| **Energy (kcal/kg/day)** | 23.03 ± 2.37[a] | 33.33 ± 4.05[a] | 24.56 ± 3.51 | **0.009** |
| **Carbohydrate (g/day)** | 169.96 ± 12.65 | 214.71 ± 21.24 | 181.06 ± 19.59 | 0.053 |
| **Protein (g/day)** | 68.92 ± 5.42[a] | 102.41 ± 13.00[a,b] | 63.26 ± 8.21[b] | **0.046** |
| **Protein (g/kg/day)** | 1.08 ± 0.10 | 1.59 ± 0.22[a] | 0.98 ± 0.12[a] | **0.041** |
| **Lipids (g/day)** | 55.13 ± 6.87[a] | 85.16 ± 12.62[a] | 56.73 ± 10.16 | **0.019** |
| Fiber (g)* | 11.95 ± 1.35[a] | 8.61 ± 0.76[a] | 10.06 ± 0.97 | **0.021** |
| Saturated fat (g)* | 12.45 ± 1.06 | 12.68 ± 1.12 | 11.38 ± 0.96 | 0.156 |
| Polyunsaturated fat (g)* | 9.75 ± 0.51 | 9.67 ± 1.11 | 9.98 ± 0.67 | 0.928 |
| Monounsaturated fat (g)* | 10.27 ± 0.52[a] | 11.24 ± 0.76[b] | 8.76 ± 0.48[a,b] | **0.001** |
| Cholesterol (mg)* | 174.00 ± 19.06[a] | 246.87 ± 28.07[a] | 111.16 ± 14.91[a] | **0.001** |
| Thiamine (mg)* | 0.84 ± 0.09[a] | 0.58 ± 0.05[a] | 0.70 ± 0.17 | **0.013** |
| Riboflavin (mg)* | 0.67 ± 0.06 | 0.51 ± 0.06 | 0.48 ± 0.07 | **0.046**[£] |
| Niacin (mg)* | 10.85 ± 1.72 | 8.27 ± 1.65 | 9.05 ± 1.43 | 0.876 |
| Vitamin B6 (mg)* | 0.42 ± 0.05 | 0.40 ± 0.10 | 0.40 ± 0.04 | 0.706 |
| Vitamin C (mg)* | 59.72 ± 11.39[a] | 66.57 ± 18.82 | 30.24 ± 6.63[a] | **0.004** |
| Sodium (mg)* | 1224.43 ± 46.15[a] | 1167.15 ± 67.58[b] | 1452.79 ± 71.59[a,b] | **0.001** |
| Iron (mg)* | 5.03 ± 0.55 | 4.80 ± 0.53 | 4.46 ± 0.34 | 0.342 |
| Calcium (mg)* | 317.56 ± 43.54 | 287.75 ± 41.02 | 289.17 ± 53.73 | 0.663 |
| Potassium (mg)* | 1940.52 ± 103.36 | 1719.79 ± 133.36 | 1782.63 ± 93.10 | 0.117 |
| Phosphorus (mg)* | 596.48 ± 40.53 | 656.48 ± 51.33[a] | 518.47 ± 38.53[a] | **0.009** |
| Magnesium (mg)* | 128.01 ± 12.53 | 132.39 ± 12.31 | 119.19 ± 9.04 | 0.445 |
| Manganese (mg)* | 1.40 ± 0.15 | 1.01 ± 0.12 | 1.34 ± 0.13 | 0.065 |
| Zinc (mg)* | 6.81 ± 0.40 | 5.99 ± 0.43 | 6.61 ± 0.31 | 0.393 |

Adjusted for sex (F/M). age (years), treatment (Surgery; RT/QT; QRT), income (SM categories) and stage (I, II, III, IV, Y_pathological).

a,b Equal letters, statistical differences between times.

* Total nutrients daily/1000kcal.

£ post hoc there was no statistical difference.

p <0.05 statistically significant value.

**Table 3. Association between colostomy time and Brazilian Healthy Eating Index score and percentage of food groups (*NOVA* classification) in colorectal cancer patients, using generalized estimation equation models.**

| Variables | T0 (n = 46) | T1 (n = 23) | T2 (n = 22) | Comparison (paired methods) | Mean difference | p-value (Bonferroni) | Observation power |
|---|---|---|---|---|---|---|---|
| | Mean ± Standard Error | | | | | | |
| **Components of the BHEI-R (score)** | | | | | | | |
| Total grains (0–5) | 4.77 ± 0.10 | 4.82 ± 0.09 | 4.91 ± 0.11 | - | - | > 0.05 | <0.80 |
| Whole grains (0–5)# | 0.59 ± 0.14 | 0.50 ± 0.18 | 0.65 ± 0.18 | - | - | > 0.05 | <0.80 |
| Total fruit (0–5)* | 2.50 ± 0.44 | 1.60 ± 0.36 | 0.12 ± 0.03 | T0-T1 | 0.89 | **0.033** | 0.68 |
| | | | | T0-T2 | 2.38 | **0.001** | **0.85** |
| | | | | T1-T2 | 1.48 | **0.001** | 0.29 |
| Whole fruit (0–5)* | 2.67 ± 0.45 | 1.25 ± 0.29 | 0.15 ± 0.04 | T0-T1 | 1.43 | **0.001** | 0.57 |
| | | | | T0-T2 | 2.54 | **0.001** | **0.91** |
| | | | | T1-T2 | 1.11 | **0.001** | 0.18 |
| Total vegetables (0–5) | 4.22 ± 0.27 | 5.11 ± 0.28 | 3.99 ± 0.45 | T0-T1 | -0.89 | **0.012** | <0.80 |
| | | | | T1-T2 | 1.12 | **0.021** | <0.80 |
| Dark green and orange vegetables and legumes (0–5)* | 3.91 ± 0.35 | 5.04 ± 0.53 | 4.14 ± 0.45 | - | - | > 0.05 | <0.80 |
| Milk and dairy products (0–10) | 4.28 ± 0.69 | 2.69 ± 0.66 | 4.17 ± 1.19 | - | - | > 0.05 | <0.80 |
| Meat, eggs and legumes (0–10) | 9.69 ± 0.24 | 9.90 ± 0.21 | 9.83 ± 0.24 | - | - | > 0.05 | <0.80 |
| Oils (0–10) | 9.54 ± 0.05 | 9.38 ± 0.07 | 9.55 ± 0.05 | - | - | > 0.05 | <0.80 |
| Saturated Fat (0–10)* | 5.41 ± 0.87 | 4.04 ± 1.06 | 6.58 ± 1.09 | - | - | > 0.05 | <0.80 |
| Sodium (0–10) | 5.95 ± 0.45 | 6.03 ± 0.52 | 4.07 ± 0.51 | T0-T2 | 1.88 | **0.001** | **0.87** |
| | | | | T1-T2 | 1.97 | **0.001** | 0.73 |
| Energy content from SoFAAS (0–20) | 17.75 ± 1.46 | 12.25 ± 2.38 | 15.65 ± 2.89 | - | - | > 0.05 | <0.80 |
| Total BHEI-R (0–100) | 73.08± 2.96 | 65.78 ± 3.50 | 66.26 ± 3.89 | T0-T2 | 6.81 | **0.022** | **0.91** |
| **Food groups (*NOVA* classification) (%)** | | | | | | | |
| Raw or minimally processed | 55.11 ± 3.46 | 42.04 ± 3.32 | 53.71 ± 3.88 | T0-T1 | 13.08 | **0.001** | **0.86** |
| | | | | T1-T2 | -11.67 | **0.001** | 0.49 |
| Culinary ingredients | 17.77 ± 1.78 | 16.96 ± 2.20 | 16.58 ± 2.33 | - | - | > 0.05 | <0.80 |
| Processed* | 6.88 ± 1.46 | 10.51 ± 2.53 | 10.21 ± 2.42 | T0-T1 | -3.63 | **0.016** | <0.80 |
| Ultra-processed | 7.93 ±1.18 | 17.75 ± 3.03 | 14.20 ± 3.68 | T0-T1 | -9.82 | **0.015** | **0.98** |

Adjusted for sex (F/M), age (years), treatment (Surgery; RT/QT; QRT), income (SM categories) and stage (I, II, III, IV, Y_pathological).

Abbreviation: BHEI-R: Brazilian Healthy Eating Index. Energy content from SoFAAS: calories from solid, saturated, and trans fats, alcohol, and added sugar. *Tweedie model

# linear model. *p* < 0.05 statistically significant value

T0-T2 (p = 0.001) and T1-T2 (p = 0.001). There was no significant difference for other components.

According to consumption by processing level, there was a reduction in the percentage of consumption of raw or minimally processed foods between T0-T1 (p = 0.001) and T0-T2 (p = 0.001). On the other hand, there was an increase in the consumption of ultra-processed foods from T0-T1 (p = 0.015) (Table 3).

## Discussion

In the present study, a decrease in diet quality was observed with respect to the consumption of total fruits (p<0.05), whole fruits (p = 0.001), and sodium (p = 0.001). Additionally, decreases were observed in the total BHEI-R score (p = 0.022) and the percentage consumption (*NOVA* classification) of raw or minimally processed foods (p = 0.001), and an increase

was observed in the ultra-processed foods consumption (p = 0.015). In addition, increases were observed in total energy consumption (p = 0.001), energy consumption per kilogram of weight (p = 0.009) and lipids (p = 0.019), decreases consumption of proteins (g/kg) (p = 0.041), fiber (p = 0.021), monounsaturated fatty acids (p = 0.001), thiamine (p = 0.013), vitamin C (p = 0.004) and phosphorus (p = 0.009).

As explained above, the total BHEI-R score decreased over time, as well as, the consumption of whole fruit component, and also the percentage of raw or minimally processed foods (*NOVA* classification). Decreased consumption of these food groups or nutrients is associated with adverse health impacts [44] because they are food sources of antioxidant vitamins, minerals, fiber, and other bioactive compounds. Additionally, some evidence has shown that these nutrients are protective factors against CRC development, reducing the risk recurrence and increasing survival in cancer survivors [4, 9, 45–47]. Therefore, it is essential to evaluate the intake of these foods as protective factors that could potentially prevent new neoplasms or recurrences, since an unhealthy diet is associated with the development of cancer [9].

These results are worrying, since during treatment, it is expected that the consumption of fruits and vegetables will be reduced at T0 due to the patient's fear of increasing the fecal volume and stimulating gut transit time, as fruits are composed of fibers and can influence stool consistency [8, 12]. The decreased consumption of fruits and vegetables can be linked mainly to fruits that have fiber in their composition and could be associated with postdischarge guidance protocol institutions. The recommendation is to reduce the intake fruit in case recurrent complications of ostomy, such of peristomal skin lesions and stoma blockage [48]. This precaution is also taken because are most recurrent complications in ostomy [49]. Some restrictions are citrus fruits these are sources of vitamins, especially vitamin C [50], and in the present study, a reduction in this vitamin was observed. In this sense, the patients probably maintain their eating habits following the same guidelines received on at the initial moment of colostomy insertion, with food intake of this group even worsening. The reduction in this consumption is also justified by the attempt to minimize prolapse risk caused by constipation, with this intestinal alteration being more common in patients with colostomy [4, 7, 8].

Regarding the *NOVA* classification, there was an increase in the percentage consumption of processed and ultra-processed foods, which leads to adverse health impacts, since these foods have a worse nutritional composition than other food groups [44]. Furthermore, there is evidence that the increase in the proportion of consumption of ultra-processed foods in the diet is related to a higher risk of cancer in general; cardiometabolic outcomes such as hypertension, diabetes, overweight and obesity; and mortality [10, 26, 46, 51]. Given these repercussions, the World Cancer Research Fund/American Institute for Cancer Research (2018) recommends limiting the consumption of processed foods to prevent cancer or post-colostomy recurrence [52]. Meyerhard *et al.* (2007) observed that the Western dietary pattern, which includes processed foods, was associated with a higher risk of colon cancer recurrence [53], corroborating the concern about maintaining a diet with a worse pattern after a colostomy.

Furthermore, an increase in sodium consumption was observed. The increase may be related to the consumption of ultra-processed foods, as this mineral is one of the main additives added to the composition of these foods [44]. This finding reinforces the concern with patients with systemic arterial hypertension found in the sample (37.5%). Even the sodium could be more in the ultra-processed, the concern about the high consumption of ultra-processed foods should be considered in nutritional guidance for CRC survivors undergoing colostomy, because it is an important guideline for the general population, even more for avoiding cancer [10].

Finally, a significant increase in energy and lipid consumption was observed between T0 and T1. In their prospective study, Bulman (2001), who evaluated the food consumption of

patients with colostomy, also observed an increase in food and lipid intake after 6 months of the postoperative period [12]. A cross-sectional study also showed that patients with ostomies consumed more than the recommended amounts of energy, carbohydrates, and lipids [13]. Initially, lower energy and lipid consumption is expected due to the effects of treatment and the immediate postoperative period, such as pain and adaptation to the colostomy [54]. Furthermore, the increase in food consumption can be influenced by psychological factors, due to stress caused by the surgery and during the ostomy adaptation process. Emotions such as stress, depression, anxiety, and social isolation, which are feelings also evident in patients with a stoma, are connected to eating choices and behavior. Generally, these conditions result in an increased consumption of unhealthy foods, rich in salt, sugar and fat, usually in large quantities and, even, in the absence of hunger [55, 56]. On the other hand, there was a reduction in total protein consumption and grams per kilogram (T1-T2). Although the mean protein intake is within the lower limit of the recommendation [57], it is crucial to observe the adequacy of protein consumption, as protein influences better metabolism and maintenance of muscle mass [57].

Although data from a specific sample were shown, these data corroborate the food consumption pattern of the general population, with an increase in the percentage of ultra-processed and refined food consumption. On the other hand, a lower frequency of consumption of raw or minimally processed foods, fruits and vegetables, and foods with lower levels of sugar and fat were also observed [58, 59].

Cancer treatment, including surgery, may influence dietary modification due to chemosensory alteration, loss of appetite, change in taste and increased sensitivity to smells, and changes in the intestinal mucosa, influencing nutrient absorption [17, 60–63]. In addition, patients in the postoperative period, have difficulties in food consumption during the adaptation process [22]. Therefore, nutritional counseling and monitoring contribute to minimizing the impact of treatment side effects, following the progression of consumption of food according to the symptomatology, ensuring an adequate dietary intake, favoring the improvement and recovery of nutritional status, helping to heal and immunity, preventing possible complications and mortality [22, 60–63].

Unfortunately, despite the huge colostomy impact in patients' life, we have not found studies with nutritional aspects besides energy and nutrient consumption. After colostomy surgery, nutrition monitoring is essential to provide knowledge for long-term sustainable self-care. Achieving eating habit changes is a challenge for the patient who feels scared about the eat some foods and trigger some uncomfortable symptom. At final, the patient has difficulty with their eating pattern self-management leading to a monotonous diet.

This study has some limitations. Initially, the 24HR was used, and although it is an accurate instrument to assess food intake, it may present a bias inherent to the method, as there is a dependence on the ability to recall food intake. However, the data collection was performed by a trained nutritionist with meticulous standardization using the Multiple Pass method. In addition, the value of consumption was considered for patients who had 24HR data from a single day. Although it probably does not represent the usual intake, the study estimates the mean patients' consumption. Therefore, the application of a single 24HR may be sufficient to estimate the mean of group intake or even the difference in this intake between more groups [32]. Finally, some outcomes did not reach the observation power of 0.8. Lower observation power increases the probability of the type II error, that is, not detecting a difference when it exists. In our results, a significant difference was found for several outcomes even with lower power of observation, probably due to the larger effect size. Thus, although it is important to detect effect size and observation power, lower observation powers are not synonymous with a lack of importance in clinical practice, especially when dealing with patients in very specific

conditions. In addition, the results with a significant difference concomitant with sufficient observation power showed consistency between the two instruments for assessing food consumption (BHEI-R and *NOVA*). Despite the follow-up losses, there was no difference in mean age, sex, tumor staging, and education level, except types of treatment between those who stayed and who withdrew from our study. The patients who remained in the study may have been submitted more frequently to chemoradiotherapy, maybe showing more significant concern and consequently more interest in the research to aid in recovery. This study also has strengths. It is noteworthy that the sample was conducted only by patients undergoing colostomy due to CRC, favoring a better analysis of food consumption, as it is a specific clinical condition. Rigorous standardization was carried out from application to quantification and classification of food items. Since it was a prospective study, the associations of food consumption analysis were possible over time points after colostomy surgery. Finally, GEE was used, which is a robust statistical analysis tool recommended in studies with this design [64, 65] that adjusts the models for potential confounders.

## Conclusion

CRC patients after colostomy have a worse diet quality, decreasing the intake of fruits and raw/minimally processed foods and increasing the intake of ultra-processed foods consumption. These findings indicate the need for better nutritional support for this population, guiding patients with a nutritional frequently appointments for a healthier diet to improve their prognosis and reduce the risk of recurrence or other chronic diseases.

Future studies only with ostomy patients by CRC with even longer-term follow-up and larger samples are needed to assess associated dietary indicators in disease recurrence risk and possible complications of colostomy.

## Supporting information

**S1 Table. Effect size and observation power of the food consumption variables of the Brazilian Healthy Eating Index and processing level (*NOVA* classification).**
(DOCX)

**S2 Table. Differences in the general characteristics of patients who remained in the study and those lost to follow-up.**
(DOCX)

## Acknowledgments

We thank Yasmin Gonçales Amaral for discussing some food groups for NOVA classification and BHEI-R scores collaborating on the initial food dataset.

## Author Contributions

**Conceptualization:** Karine de Almeida Silva, Geórgia das Graças Pena.

**Data curation:** Arenamoline Xavier Duarte, Karine de Almeida Silva.

**Formal analysis:** Arenamoline Xavier Duarte, Isabela Borges Ferreira, Cristiana Araújo Gontijo, Geórgia das Graças Pena.

**Methodology:** Geórgia das Graças Pena.

**Project administration:** Geórgia das Graças Pena.

**Writing – original draft:** Arenamoline Xavier Duarte.

**Writing – review & editing:** Arenamoline Xavier Duarte, Karine de Almeida Silva, Isabela Borges Ferreira, Cristiana Araújo Gontijo, Geórgia das Graças Pena.

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
