## [Decision Letter · Decision Letter 0]

4 Jul 2024

PONE-D-24-06878Increased consumption of ultra-processed foods and worse diet quality in colorectal cancer patients after colostomy: A prospective studyPLOS ONE

Dear Dr. Pena,

Thank you for submitting your manuscript to PLOS ONE. After careful consideration, we feel that it has merit but does not fully meet PLOS ONE’s publication criteria as it currently stands. Therefore, we invite you to submit a revised version of the manuscript that addresses the points raised during the review process.

We look forward to receiving your revised manuscript.

Kind regards,

Stanisław Jacek Wroński, M.D., Ph.D, FEBU

Academic Editor

PLOS ONE

Journal Requirements:

3. We notice that your supplementary tables are included in the manuscript file. Please remove them and upload them with the file type 'Supporting Information'. Please ensure that each Supporting Information file has a legend listed in the manuscript after the references list.

Additional Editor Comments:

Please find the following comments on the manuscript "Increased consumption of ultra-processed foods and worse diet quality in colorectal cancer patients after colostomy: A prospective study" made by the reviewers

Reviewer 1

improvements in abstract section by incorporating key outcomes of the study. Please add latest references in introduction section and in discussion to validate the results. The conclusion section can be further improved by mentioning the future avenue for the current research.

Reviewer 2

An issue of appropriate management of patients with an intestinal stoma was rightly highlighted by the authors.

In my opinion, this should be added:

1. The issue of psychological conditions that may be the basis of malnutrition such as 'stress eating' related to the disease, the treatment and the existing stoma.

2. Explain in more detail the issue of type II error and why this parameter was used in this study.

3. Does the study take into account the socio-economic status of the patients? Certainly education and level of intelligence influence the health-seeking behaviour of patients.

The above comments do not undermine my positive opinion. I believe that after the above mentioned corrections/additions the paper will be suitable for publication in PLOS ONE.

I kindly ask you to take note of the above comments and to address them in your work accordingly

sincerely

Stanisław Wroński

Academic Editor

Reviewers' comments:

Reviewer's Responses to Questions

**Comments to the Author**

1. Is the manuscript technically sound, and do the data support the conclusions?

Reviewer #1: Yes

Reviewer #2: Yes

2. Has the statistical analysis been performed appropriately and rigorously? 

Reviewer #1: Yes

Reviewer #2: Yes

3. Have the authors made all data underlying the findings in their manuscript fully available?

Reviewer #1: Yes

Reviewer #2: Yes

4. Is the manuscript presented in an intelligible fashion and written in standard English?

Reviewer #1: Yes

Reviewer #2: Yes

5. Review Comments to the Author

Reviewer #1: The manuscript needs some improvements in abstract section by incorporating key outcomes of the study. Please add latest references in introduction section and in discussion to validate the results. The conclusion section can be further improved by mentioning the future avenue for the current research.

Reviewer #2: An issue of appropriate management of patients with an intestinal stoma was rightly highlighted by the authors.

In my opinion, this should be added:

1. The issue of psychological conditions that may be the basis of malnutrition such as 'stress eating' related to the disease, the treatment and the existing stoma.

2. Explain in more detail the issue of type II error and why this parameter was used in this study.

3. Does the study take into account the socio-economic status of the patients? Certainly education and level of intelligence influence the health-seeking behaviour of patients.

The above comments do not undermine my positive opinion. I believe that after the above mentioned corrections/additions the paper will be suitable for publication in PLOS ONE.

6. PLOS authors have the option to publish the peer review history of their article (what does this mean?). If published, this will include your full peer review and any attached files.

Reviewer #1: **Yes: **Dr. Aftab Ahmed

Reviewer #2: No

---

## [Author Response · Author response to Decision Letter 0]

18 Aug 2024

Uberlândia, Brazil

August 18th, 2024

Stanisław Wroński

Academic Editor

PLOS ONE

Dear Editor,

Thank you for the opportunity to submit the revision of PONE-D-24-06878 “Increased consumption of ultra-processed foods and worse diet quality in colorectal cancer patients after colostomy: A prospective study”. We appreciate the comments and suggestions presented by the reviewers. They highlighted important issues and their inputs are extremely helpful for improving the manuscript. We highlighted the changes in our manuscript in red and enumerated all points raised by the reviewers with a response for each one of them. We hope that the reviewers will find our responses to their comments satisfactory, and we are willing to finish the revised version of the manuscript including any further suggestions that the reviewers may have.

Yours sincerely,

Geórgia das Graças Pena

Graduate Program in Health Sciences

Federal University of Uberlandia, Uberlandia, Minas Gerais, Brazil

E-mail: georgia@ufu.br

ANSWERS TO THE REVIEWER’S COMMENTS

(The responses are in red letter immediately below the reviewer’s comments)

Journal Requirements

https://journals.plos.org/plosone/s/file?id=wjVg/PLOSOne_formatting_sample_main_body.pdf and https://journals.plos.org/plosone/s/file?id=ba62/PLOSOne _formatting_sample_title_authors_affiliations.pdf

Comment to Editor: All Journal’s requirements were revised and the manuscript is corrected now.

Comment to Editor: thank you for pointing this out. We are sending the complete dataset including all variables used in our results. In addition, we have corrected some terms in Portuguese in the data set.

3. We notice that your supplementary tables are included in the manuscript file. Please remove them and upload them with the file type 'Supporting Information'. Please ensure that each Supporting Information file has a legend listed in the manuscript after the references list.

Comment to Editor: We appreciate Reviewer's comment. We revised the manuscript and adjusted it, as requested. 

Comment to Editor: thank you for your comment, we reviewed the references and did not find any retractions of the articles cited. However, the previous reference was deleted (Yoshikawa R. Structure and composition of skin barriers. Skin barriers for stoma care: from basic theory to clinical application. Alcare. 2000:17-23). We added some missing data that we noticed after the conference, the manuscript was adjusted and now the references are complete and correct.

Additional Editor Comments:

Reviewer 1

Improvements in abstract section by incorporating key outcomes of the study. Please add latest references in introduction section and in discussion to validate the results. The conclusion section can be further improved by mentioning the future avenue for the current research.

Comment to Reviewer#1: thank you for your valuable feedback on our manuscript. We appreciate your comments and the opportunity to address your concerns regarding sections in the Abstract, Introduction, Discussion, and Conclusion sections. We revised the manuscript and included more references about colorectal cancer and ostomy, as requested. However, studies about ostomy with colorectal cancer patients are very scarce in the literature. So, we have included the following sentences below.

• Clean version: Page,3, lines 57-60

Ostomy due to changes in intestinal function requires some management during the different adaptations, considering the underlying pathology, and one of the management options is dietary change, which aims to control intestinal function before some gastrointestinal symptoms. [4-6].

• Clean version, Page,20, lines 349-354

Decreased consumption of these food groups or nutrients is associated with adverse health impacts [44] because they are food sources of antioxidant vitamins, minerals, fiber, and other bioactive compounds; additionally, evidence has shown that these nutrients are protective factors against CRC development, reduce risk recurrence and increased survival in cancer survivors [4,9,45-47].

• Clean version, Page 21, lines 367-369

The recommendation is to reduce the intake of citrus fruit in case of peristomal skin lesions since the pectin contained in this food in contact with the water present in the feces can acidify the environment [48]. This precaution is also taken because this lesion is one of the most recurrent complications in ostomy [49]. Citrus fruits are a source of vitamins, especially vitamin C [50], and in the present study, a reduction in this vitamin was observed.

• Clean version, Page 22, lines 402-403

A cross-sectional study also showed that patients with ostomies consumed more than the recommended amounts of energy, carbohydrates, and lipids [13].

• Clean version, Page 22, lines 406-412

Furthermore, the increase in food consumption can be influenced by psychological factors, due to stress caused by the surgery and during the ostomy adaptation process. Emotions such as stress, depression, anxiety, and social isolation, which are feelings also evident in patients with a stoma, are connected to eating choices and behavior. Generally, these conditions result in an increased consumption of unhealthy foods, rich in salt, sugar and fat, usually in large quantities and, even, in the absence of hunger [55,56].

• Clean version, Page 25, lines 482-484

Future studies only with ostomy patients by CRC with even longer-term follow-up and larger samples are needed to assess associated dietary indicators in disease recurrence risk and possible complications of colostomy.

Reviewer 2

An issue of appropriate management of patients with an intestinal stoma was rightly highlighted by the authors.

In my opinion, this should be added:

1. The issue of psychological conditions that may be the basis of malnutrition such as 'stress eating' related to the disease, the treatment and the existing stoma.

Comment to Reviewer#2: We understand and agree with the Reviewer’s concern. We revised the manuscript and included this information, as requested. So, we have included the following sentences below.

• Clean version, Page 22, lines 406-412

Furthermore, the increase in food consumption can be influenced by psychological factors, due to stress caused by the surgery and during the ostomy adaptation process. Emotions such as stress, depression, anxiety, and social isolation, which are feelings also evident in patients with a stoma, are connected to eating choices and behavior. Generally, these conditions result in an increased consumption of unhealthy foods, rich in salt, sugar, and fat, usually in large quantities and, even, in the absence of hunger [55,56].

References

• Stavropoulou A, Vlamakis D, Kaba E, Kalemikerakis I, Polikandrioti M, Fasoi G, et al. “Living with a Stoma”: Exploring the Lived Experience of Patients with Permanent Colostomy. International Journal of Environmental Research and Public Health. 2021 Jan 1;18(16):8512. https://doi.org/10.1371/journal.pone.0302914.

• Ljubičić M, Matek Sarić M, Klarin I, Rumbak I, Colić Barić I, Ranilović J, et al. Emotions and Food Consumption: Emotional Eating Behavior in a European Population. Foods [Internet]. 2023 Jan 1;12(4):872.https://doi.org/10.3390/foods12040872

2. Explain in more detail the issue of type II error and why this parameter was used in this study.

Comment to Reviewer#2: Thank you for bringing this important point. We understand your concern about the issue of type II error. Exploring the subject, it is important to emphasize the method of statistical analysis Generalized Estimation Equations (GEE) is the most appropriate method to proceed with longitudinal analyses which the estimative of models considering the patient is his own control, being compared to himself over time (Wang et al 2013). When the analysis is restricted to complete cases and missing data depends on previous responses, the Generalized Estimating Equation (GEE) approach, commonly used when population-averaged effects are of primary interest, may lead to biased parameter estimates. The GEE procedure provides a weighted method for analyzing longitudinal data with missing observations, thereby extending the standard Generalized Estimating Equations. We believe this approach has minimized the impact of missing data.

GEE can greatly enable better comparisons between groups by controlling variation for additional data from single participant measurements in a data set (Friedel et al, 2019). 

To minimize the risk of confusion in the analysis, declaring that there is no association when the truth is limited to type II error is important, this type of error is present in more than 20% of studies. Loss of patient follow-up is one of the main causes of bias leading to type II error (Domb et al, 2021). To reduce the effect of error, it is important to consider the application of the power of a statistical test in the analysis, but this limitation is associated with the sample size. According to research, it can be concluded that the power with a small sample is the effect size for larger, the real effect is considered by researchers to be a power of 80% or more (Shreffle & Huecker, 2023). The present study, all these precautions were taken from the application to the interpretation of the analyses.

References

• Wang M, Kong L, Li Z, Zhang L. Covariance estimators for generalized estimating equations (GEE) in longitudinal analysis with small samples. Stat Med. 2016;35(10):1706-1721. doi:10.1002/sim.6817.

• Friedel JE, DeHart WB, Foreman AM, Andrew ME. A Monte Carlo method for comparing generalized estimating equations to conventional statistical techniques for discounting data. Journal of the Experimental Analysis of Behavior. 2019 Jan 24;111(2):207–24. doi:10.1002/jeab.497

• Shreffler J, Huecker MR. Type I and Type II Errors and Statistical Power [Internet]. PubMed. Treasure Island (FL): StatPearls Publishing; 2023.

• Domb BG, Sabetian PW. The Blight of the Type II Error: When No Difference Does Not Mean No Difference. Arthroscopy: The Journal of Arthroscopic & Related Surgery. 2021 Apr;37(4):1353–6. doi:10.1016/j.arthro.2021.01.057

Therefore, we added the S1 and S2 tables as supplementary material. To clarify this point, So, we have the following sentences below.

• Clean version, Page 24, lines 451-460

Lower observation power increases the probability of the type II error, that is, not detecting a difference when it exists. In our results, a significant difference was found for several outcomes even with lower power of observation, probably due to the larger effect size. Thus, although it is important to detect effect size and observation power, lower observation powers are not synonymous with a lack of importance in clinical practice, especially when dealing with patients in very specific conditions. In addition, the results with a significant difference concomitant with sufficient observation power showed consistency between the two instruments for assessing food consumption (BHEI-R and NOVA).

3. Does the study take into account the socio-economic status of the patients? Certainly education and level of intelligence influence the health-seeking behaviour of patients.

Comment to Reviewer#2: We appreciate your attention to detail regarding the socio-economic status of the patients and we understand your concern. However, the income could be considered a proxy of the socioeconomic conditions, i.e., one is expected to impact the other. This consideration is important to avoid collinearity between variables in the models. Both inclusions could confuse the results, or yet, an overload of them. Even so, we performed some analyses, since socioeconomic conditions also could be important. When Kruskal-Wallis tests were conducted, we observed there weren’t differences between education level with the BHEI-R components or NOVA variables for the majority of the variables. Only “Total vegetables”, “Dark green and orange vegetables, and legumes” (BHEI-R components), and “percentage raw” (NOVA classifications) showed differences between education level groups. Besides that, as a low-income country, the family income could be represented better in the health-seeking behavior of patients in Brazil than education level. Therefore, following statistical recommendations to avoid collinearity, the lack of differences between education level and the main outcomes, and the Brazil education context, we choose to maintain the family income in the models to represent better the manuscript results.

---

## [Decision Letter · Decision Letter 1]

29 Aug 2024

Increased consumption of ultra-processed foods and worse diet quality in colorectal cancer patients after colostomy: A prospective study

PONE-D-24-06878R1

Dear Dr. Georgia das Gracas Pena

We’re pleased to inform you that your manuscript has been judged scientifically suitable for publication and will be formally accepted for publication once it meets all outstanding technical requirements.

Kind regards,

Stanisław Jacek Wroński, M.D., Ph.D, FEBU

Academic Editor

PLOS ONE

---

## [Editor Report · Acceptance letter]

6 Nov 2024

PONE-D-24-06878R1 

PLOS ONE

Dear Dr. Pena, 

I'm pleased to inform you that your manuscript has been deemed suitable for publication in PLOS ONE. Congratulations! Your manuscript is now being handed over to our production team.

Kind regards, 

on behalf of

Dr. Stanisław Jacek Wroński 

Academic Editor

PLOS ONE